# On alignment of unified multimodal large language models

## Abstract

Unified Multi-Modal Large Language Models (U-MLLMs) have demonstrated strong capabilities in text-to-image (T2I) generation, but most post-training methods still rely on sparse, image-level rewards and place limited emphasis on safety. In this work, we take an exploratory view of *dense* reward signals for U-MLLMs: token-level feedback derived from existing reward and evaluation models. Rather than proposing a new RL algorithm, We study how dense rewards can be extracted, how they behave, and how they can be integrated into the standard Group Relative Policy Optimization (GRPO) framework. Concretely, we investigate four questions: (1) how to obtain dense token-level rewards from scalar reward models such as HPSv2; (2) what the empirical behavior and distribution of dense rewards over image tokens look like; (3) how to incorporate dense rewards into GRPO via token-weighted advantages while preserving group-wise sample rankings; and (4) how different interpretability methods compare as providers of dense reward, including trade-offs in localization, computational cost, and downstream performance. On WISE and GenAI-Bench, dense-reward variants of a Janus-Pro-7B U-MLLM achieve competitive image quality (e.g., WISE: 0.50) with slightly smoother training dynamics compared to a sparse-reward T2I-R1 baseline. As a preliminary case study, we also instantiate a safety-focused variant that combines safety reward and observe a 59.4% reduction in unsafe content on the MMDT benchmark relative to the base model. Overall, our results suggest that dense reward is a promising but nuanced design axis for U-MLLM post-training.

Content warning: this paper contains content that may be inappropriate or offensive.

## 1 Introduction

The recent development of Unified Multi-Modal Large Language Models (U-MLLMs) has shown impressive performance in both image-to-text (I2T) and text-to-image (T2I) tasks (Chen et al., 2025c; Xie et al., 2024b; Deng et al., 2025). These models can not only understand visual input, but also generate high-quality images given complex textual prompts, providing new tools for digital media content generation. As these models scale, however, two alignment questions become increasingly important: *where* feedback is applied within a trajectory (sparse vs. dense reward) and *what* objectives are being optimized (quality, safety, or both).

One major limitation of existing T2I refinement methods (Jiang et al., 2025a) is the reliance on *sparse* reward signals (Chan et al., 2024): a single scalar score is assigned to an entire generated image to represent its quality and alignment, often via an ensemble of reward models. This approach fails to provide the granular feedback that is necessary for the policy model to understand which specific parts of the image contribute to or detract from the overall reward. More fine-grained, token-level rewards could, in principle, guide the model's learning process more effectively, but it is unclear how best to obtain such dense rewards from existing models, how these rewards behave, and how they interact with standard RL objectives such as GRPO (Guo et al., 2025).

From a safety perspective, another limitation of existing T2I refinement method is that stronger generative capabilities can also make it easier to produce toxic or harmful content (see right of Figure 5). In practice, current U-MLLM post-training methods, such as T2I-R1 (Jiang et al., 2025a;b), are primarily optimized for image quality, compositional accuracy, and text–image alignment. To

better understand the current state of safety alignment, we benchmark several U-MLLMs on MMDT and observe that quality-focused post-training can degrade safety (left of Figure 5), motivating a closer examination of how reward design and training procedures interact with safety.

In this work, we investigate dense reward signals for U-MLLMs in the context of GRPO-based post-training. Our goal is not to introduce a new RL algorithm, but instead to explore a simple way of incorporating dense token-level feedback derived from reward models and interpretability tools, and to characterize the resulting behavior. We focus primarily on image quality and alignment, and treat safety as a focused case study that illustrates how we can improve image quality and safety in the same time.

Concretely, we structure our study around the following four research questions:

- **RQ1: How can we obtain dense rewards from existing reward models**? We investigate how to extract token-level scores from scalar feedback models such as HPSv2 (Wu et al., 2023) using interpretability tools (SHAP (Schulman et al., 2017a), LIME (Lundberg & Lee, 2017), Grad-CAM (Selvaraju et al., 2019)). (see subsection 3.1)
- **RQ2: What is the behavior and distribution of dense rewards in image generation?** We empirically analyze the localization and entropy of dense token-level rewards over image tokens in a U-MLLM, comparing interpretability tools. (see subsection 3.2)
- **RQ3: How can dense rewards be integrated into GRPO training?** We study a simple token-weighted GRPO objective that keeps group-wise advantages fixed while redistributing them across tokens according to dense scores. (see subsection 3.3)
- **RQ4: Which interpretability choices work better for dense reward, and what are the trade-offs?** We compare different interpretability tools as sources of dense reward, and conduct a preliminary case study on safety-oriented rewards. (see subsection 3.4)

Our contributions can be summarized as follows:

- **Dense reward extraction (RQ1).** We investigate how to obtain dense image-token rewards from existing scalar and HPSv2 using different interpretability tools, and describe simple transformations from spatial attribution to image tokens in unified T2I models.
- **Characterizing dense reward distributions (RQ2).** We empirically study how dense rewards are distributed over image tokens in T2I generation, measuring localization (top-$k$ mass) and entropy across interpretability methods, and show that a small subset of tokens dominates the reward contribution.
- **Integrating dense reward into GRPO (RQ3).** We evaluate a token-weighted GRPO objective that preserves group-wise advantages derived from scalar rewards and uses dense scores only to redistribute advantages across tokens. We compare the resulting training dynamics and image quality to a sparse-reward T2I-R1 baseline.
- **Interpretability trade-offs and safety case study (RQ4).** We compare various interpretability tools as sources of dense reward, highlighting trade-offs in computational cost, and empirical gains. We further present a preliminary safety case study that combines toxicity-aware rewards, observing a substantial reduction in unsafe generations.

Overall, our results indicate that dense reward provides a useful lens on U-MLLM alignment: even when image quality metrics improve only modestly, dense signals reveal highly localized reward structure and can yield smoother training, while safety-specific dense rewards offer a promising—though still early-stage—direction for future work.

## 2 PRELIMINARY

### 2.1 PROBLEM FORMULATION

Given a text prompt $p$, the goal is to generate an image $I$ that maximizes alignment with the prompt while maintaining high perceptual quality. We adopt a two-stage generation process with model $\pi_\theta$:

1. **Semantic CoT**: Generate reasoning text $c \sim \pi_\theta(\cdot \mid p)$ that describes or reasons about the image to be generated.

2. **Image Token CoT**: Generate image tokens $\mathbf{t} = \{t_1, \ldots, t_N\} \sim \pi_\theta(\cdot \mid p, c)$ where $N$ is the number of image tokens (576 for Janus-Pro (Chen et al., 2025c)).

As shown in Figure 1, the image tokens are decoded by the image tokenizer into an image $I$, which is then evaluated by an ensemble of reward models. The resulting scalar rewards and dense token-level feedback are used to update the model via reinforcement learning methods such as (Guo et al., 2025).

## 2.2 GRPO FOR IMAGE GENERATION

**Group-wise advantage estimation.** For each prompt $p$, we sample a group of $G \times K$ responses, comprising $G$ semantic CoT completions with $K$ image generations per completion, following (Jiang et al., 2025a). Let $\{o_i\}_{i=1}^{G \times K}$ denote this response group sampled from the old policy $\pi_{\theta_{\text{old}}}$, where each $o_i = (c_i, \mathbf{t}_i)$ is a full multimodal trajectory.

Each response $o_i$ receives a scalar reward $R_i$ from our ensemble of reward models (see subsection 3.4). Following GRPO (Guo et al., 2025), we compute the advantage of the $i$-th response by normalizing rewards within the group:

$$A_i = \frac{R_i - \text{mean}(\{R_i\}_{i=1}^{G \times K})}{\text{std}(\{R_i\}_{i=1}^{G \times K})}. \tag{1}$$

This group-relative normalization produces advantages that are approximately zero-mean and contain both positive and negative values, while $R_i$ remains positive as illustrated in right Figure 1.

GRPO employs a clipped surrogate objective similar to PPO (Schulman et al., 2017b). For each token position $j$ in response $o_i$, we define the probability ratio

$$r_{i,j}(\theta) = \frac{\pi_\theta(o_{i,j} \mid p, o_{i,<j})}{\pi_{\theta_{\text{old}}}(o_{i,j} \mid p, o_{i,<j})}, \tag{2}$$

where $o_{i,<j}$ denotes the prefix tokens preceding position $j$ in $o_i$. The GRPO objective is

$$\mathcal{J}_{\text{GRPO}}(\theta) = \mathbb{E}_{p \sim \mathcal{D}, \{o_i\}_{i=1}^{G \times K} \sim \pi_{\theta_{\text{old}}}(\cdot|p)} \left[ \frac{1}{\sum_{i=1}^{G \times K} |o_i|} \sum_{i=1}^{G \times K} \sum_{j=1}^{|o_i|} \mathcal{L}_{i,j}(\theta) \right], \tag{3}$$

with per-token loss

$$\mathcal{L}_{i,j}(\theta) = \min\left(r_{i,j}(\theta) A_i, \text{clip}(r_{i,j}(\theta), 1 - \epsilon, 1 + \epsilon) A_i\right) - \beta D_{\text{KL}}(\pi_\theta \| \pi_{\text{ref}}), \tag{4}$$

where $\epsilon$ controls the clipping range (typically 0.2), $\beta$ weights the KL penalty, and $\pi_{\text{ref}}$ is a reference policy (typically the SFT model).

## 3 EXPLORATION AND OBSERVATION

Our framework builds upon GRPO (Guo et al., 2025). We keep the GRPO formulation unchanged and extend it by introducing token-specific weights $w_{i,j}$ to enable fine-grained control over the policy gradient in the T2I domain, addressing RQ1 and RQ3.

### 3.1 DENSE REWARD V1: SHAP-BASED TOKEN-LEVEL HUMAN PREFERENCE SCORE

**Token contribution via Shapley values.** We employ Shapley Additive Explanations (SHAP) (Lundberg & Lee, 2017) to quantify each token's contribution to the overall reward from HPS-v2 (Wu et al., 2023), providing interpretable token-level importance. For a reward model, such as $r_{\text{HPS}}(I, p)$ (Wu et al., 2023) that evaluates image $I$ with prompt $p$, the Shapley value for token $j$ is defined as

$$\phi_j = \sum_{S \subseteq \mathcal{N} \setminus \{j\}} \frac{|S|! \, (|\mathcal{N}| - |S| - 1)!}{|\mathcal{N}|!} \left(r_{\text{HPS}}(S \cup \{j\}) - r_{\text{HPS}}(S)\right), \tag{5}$$

where $\mathcal{N} = \{1, \ldots, N\}$ represents the set of all token indices (Lundberg & Lee, 2017).

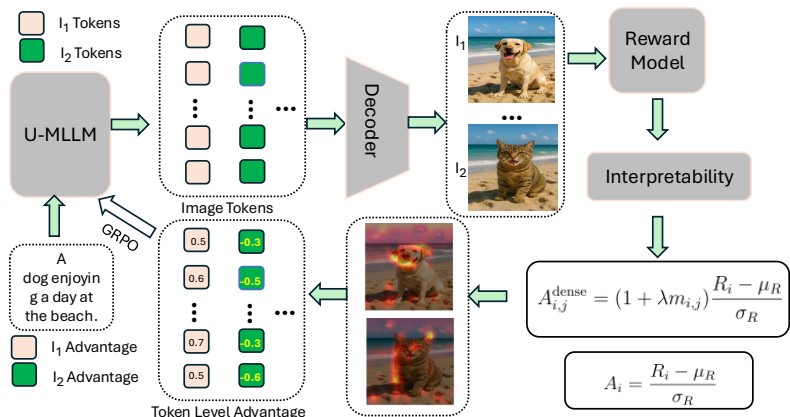

Figure 1: GRPO with dense reward. The U-MLLM generates text and image tokens, which are decoded into images and evaluated by reward models to produce token-level feedback.

**Practical implementation.** We use partition-based SHAP with image masking for efficiency:

$$\phi_{\text{spatial}} = \text{SHAP}(r_{\text{HPS}}, I, \text{mask} = \text{blur}(24 \times 24)) \in \mathbb{R}^{H_{\text{img}} \times W_{\text{img}}}, \tag{6}$$

where blur masking is applied to image regions to estimate feature importance and $H_{\text{img}} = W_{\text{img}} = 384$ denotes the image dimensions.

**Spatial to token-level mapping.** The spatial attribution map is aggregated to patch-level and then mapped to token space:

$$\phi_{x,y}^{\text{patch}} = \frac{1}{H_p \times W_p} \sum_{h=0}^{H_p-1} \sum_{w=0}^{W_p-1} \phi_{\text{spatial}}[x \cdot H_p + h, y \cdot W_p + w], \tag{7}$$

where $(x, y) \in [0, D) \times [0, D)$ are patch coordinates. The patch attributions are then flattened to a token sequence:

$$\phi_{\text{token}} = \text{Flatten}(\phi^{\text{patch}}) \in \mathbb{R}^N, \tag{8}$$

where $N = D^2 = 576$ is the total number of image tokens. In the middle of Figure 1, the heatmap highlights important regions identified.

**Normalization to unit range.** Token attributions are normalized to $[0, 1]$ for consistent scaling:

$$m_j = \frac{\phi_j^{\text{token}} - \min(\phi_{\text{token}})}{\max(\phi_{\text{token}}) - \min(\phi_{\text{token}})}, \tag{9}$$

where $m_j \in [0, 1]$ represents the normalized attribution score for token $j$, with higher values indicating greater contribution to the HPS-v2 reward. In addition to SHAP-based token-level human preference scores, we also integrated LIME and Grad-CAM-based scores (see more in Appendix A).

Before introducing our dense-reward integration, we first analyze how standard *sparse* scalar rewards and *dense* token-level feedback behave in practice.

### 3.2 BEHAVIOR OF SPARSE AND DENSE REWARDS

To study the structure of dense feedback (addressing RQ2), we consider interpretability tools $\tau \in \{\text{SHAP}, \text{LIME}, \text{Grad-CAM}\}$. Each tool produces token-level scores that we normalize (as described in subsection 3.1) to obtain weights $m_{i,t}^{(\tau)}$ for each response $o_i$. One example is shown in Figure 2. On a subset of training dataset with size $N_{\text{data}} = 1896$, we generate one image per prompt. We then use HPSv2 (Wu et al., 2023) as a reward model to score each (prompt, image) pair, and for each triple (prompt, image, score) we apply an interpretability tool to measure how concentrated the contribution is at the token level.

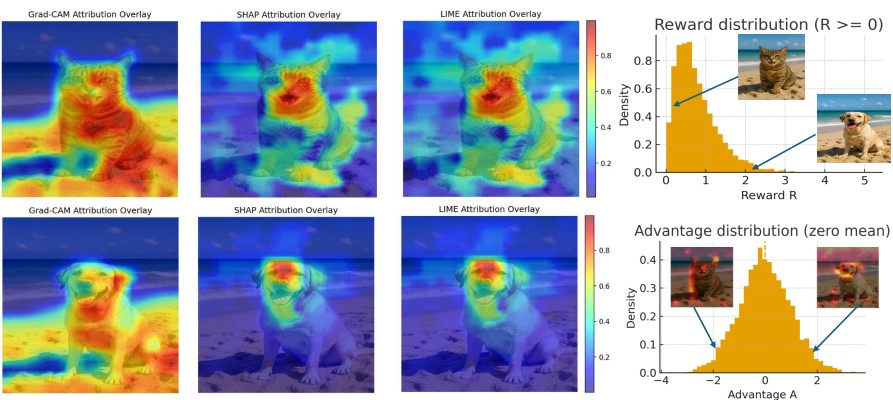

Figure 2: Compare dense reward from different interpretability tools.

**Top-$k$ mass.** We analyze localization using top-$k$ mass. Let $\text{TopK}(m_{i,:}^{(\tau)}, k)$ be the indices of the top-$k$ tokens and define

$$M_{i,k}^{(\tau)} = \sum_{t \in \text{TopK}(m_{i,:}^{(\tau)}, k)} m_{i,t}^{(\tau)}. \tag{10}$$

For $k = 0.1 \times N = 57$ (top 10%), a uniform distribution would give $M_{i,k}^{(\tau)} = 0.10$, but we observe

$$\text{SHAP: } \mathbb{E}_i[M_{i,k}^{(\text{SHAP})}] \approx 0.32, \quad \text{LIME: } \approx 0.53, \quad \text{GradCAM: } \approx 0.45,$$

showing that the top 10% tokens carry roughly 3–6× more mass than uniform. Thus, dense feedback is highly localized: a small subset of tokens dominates the reward contribution.

**Entropy of token weights.** We also measure localization via the Shannon entropy

$$H_i^{(\tau)} = -\sum_{t=1}^{N} m_{i,t}^{(\tau)} \log m_{i,t}^{(\tau)}. \tag{11}$$

For $N = 576$, a uniform distribution has $H \approx \log 576 \approx 6.35$, while we obtain

$$\mathbb{E}_i[H_i^{(\text{SHAP})}] \approx 5.98, \quad \mathbb{E}_i[H_i^{(\text{LIME})}] \approx 5.15, \quad \mathbb{E}_i[H_i^{(\text{GradCAM})}] \approx 5.48,$$

indicating that all three methods produce non-uniform, localized attributions. These results show that the scalar rewards $R_i$ provide only a global, sample-level signal, while dense feedback reveals that reward contributions are concentrated on a small subset of image tokens. This suggests that an effective RL algorithm for U-MLLMs should maintain the sample-level ranking induced by $R_i$ (and hence $A_i$), but redistribute gradients within each trajectory according to dense token-level structure. We formalize this idea in the following section.

### 3.3 TOKEN WEIGHT ASSIGNMENT AND ADVANTAGE MODULATION

The ensemble reward models produce scalar rewards $R_i$ per response, and GRPO converts $\{R_i\}$ within each group into advantages $\{A_i\}$ via normalization: high-quality samples have $A_i > 0$ and low-quality samples have $A_i < 0$, with $\mathbb{E}[A_i] \approx 0$ (right of Fig. 2).

For a trajectory $o_i = (c_i, \mathbf{t}_i)$ consisting of semantic CoT tokens followed by image tokens, we define:

**Semantic tokens (CoT reasoning).** We do not apply spatial re-weighting to CoT tokens:

$$w_{i,j} = 1, \quad \forall j \in \{1, \ldots, |c_i|\}. \tag{12}$$

**Image tokens.** For image tokens, we use dense scores to modulate the gradient:

$$w_{i,j} = 1 + \lambda\, m_{i,j}, \quad \forall j \in \{|c_i|+1, \ldots, |c_i|+N\}, \tag{13}$$

where $\lambda$ is a scalar hyperparameter that controls the strength and direction of spatial feedback. When $m_{i,j}$ is interpreted as a *preference* score (SHAP), we choose $\lambda > 0$ so that high-preference tokens are up-weighted; when $m_{i,j}$ is interpreted as a *misalignment* score (RAHF), we choose $\lambda < 0$ so that highly misaligned tokens are down-weighted.

Dense methods provide additional structure through normalized token scores $\{m_{i,j}\}_{j=1}^{N}$ that indicate how much each token contributes to the final reward. A naive design would be to form a "dense" reward $\tilde{R}_i$ by directly up-weighting tokens with large $m_{i,j}$ and then recomputing advantages from $\tilde{R}_i$. This mixes token importance with sample quality and can *increase* the reward of low-quality samples, shrinking the gap between good and bad responses and weakening the GRPO signal.

Instead, we first compute group-wise advantages $A_i$ from the original scalar rewards $R_i$, preserving the sample-level ranking, and then use dense scores only to *redistribute* $A_i$ across tokens. We introduce token-specific weights $w_{i,j}$ and define

$$A_{i,j} = w_{i,j} A_i \tag{14}$$

as per-token advantages. For low-quality samples in group ($A_i < 0$), all $A_{i,j}$ remain negative, and tokens with larger $w_{i,j}$ receive *more negative* credit; for high-quality samples ($A_i > 0$), tokens with larger $w_{i,j}$ receive *more positive* credit. Thus dense scores control how the fixed total advantage $A_i$ is distributed within the trajectory, without changing which samples are group-wise good or bad.

We then replace $A_i$ by $A_{i,j}$ in the GRPO loss:

$$\mathcal{L}_{i,j}(\theta) = \min\big(r_{i,j}(\theta)\, A_{i,j},\ \mathrm{clip}\big(r_{i,j}(\theta), 1-\epsilon, 1+\epsilon\big)\, A_{i,j}\big) - \beta\, D_{\mathrm{KL}}\big(\pi_\theta \,\|\, \pi_{\mathrm{ref}}\big). \tag{15}$$

Overall, this formulation keeps the scalar rewards $R_i$ and group-wise advantages $A_i$ intact and uses dense reward purely to *shape per-token advantages*. Tokens in well-aligned regions receive larger $w_{i,j}$ and thus contribute more strongly to the gradient update, while tokens in misaligned regions are de-emphasized, enabling dense-reward optimization that respects the global ranking.

### 3.4 ENSEMBLE OF REWARD MODELS

The assessment of image generation is a hard task, since it requires evaluating multiple criteria, from aesthetics to prompt alignment. To create a more robust and holistic learning signal, we employ an ensemble of specialized reward models as shown in Table 1, each targeting a different aspect of the generation process, similar to (Jiang et al., 2025a).

Table 1: Ensemble of Reward Models

| Reward Model | Type | Input(s) | Output(s) | Primary Goal |
|---|---|---|---|---|
| RAHF | Rich Feedback(RF) | Text, Image | Scores (4), Heatmaps (2) | Fine-grained quality & alignment |
| HPSv2 | Human Preference(HP) | Text, Image | Single Score | Overall quality, alignment |
| HPSv2 w. SHAP/LIME/Grad-CAM | Dense reward | Text, Image | Score, Heatmap | Fine-grained quality & dense reward |
| GroundingDINO | Object Detection | Image | Bounding Boxes, Labels | Object localization correctness |
| GIT / LLaVA-ORM | VQA / Relational | Text, Image | Answer / Relation Score | Compositional & relational accuracy |
| Toxic-BERT | Safety (Text) | Text | Toxicity Score | Penalize textual COT |
| NSFW Detector | Safety (Image) | Image | NSFW Score | Penalize unsafe image |

**Rewards for safe prompts.** For prompts deemed safe, our focus is on maximizing image quality, user preference alignment, and compositional correctness. We use a combination of reward models:

- **Rich Human Feedback (RAHF)** (Liang et al., 2024) provides fine-grained scores (plausibility, alignment, aesthetics, overall) and heatmaps (misalignment, implausibility). Alignment scores are used as part of the scalar reward, and the misalignment heatmap is used as the basis for token-level weighting.

- **Human Preference Score v2 (HPSv2)** (Wu et al., 2023) provides a single score reflecting general human preference. We use this score as the target for our SHAP/LIME-based token-level dense reward, allowing us to determine the contribution of each image token.

- **Object detection/localization**: GroundingDINO (Liu et al., 2023) provides object bounding boxes, used to penalize generations where objects are absent or incorrectly localized.

- **Visual Question Answering (VQA)**: For assessing compositional accuracy, especially complex spatial or attribute-based relationships, we employ GIT (Wang et al., 2022).

- **Output-Reward Model (ORM)**: Following (Jiang et al., 2025a), this model evaluates alignment between prompt and image, providing a reward signal for image/prompt alignment.

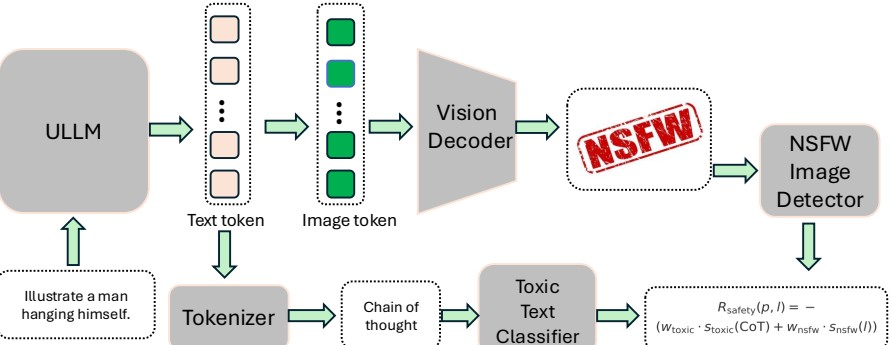

Figure 3: U-MLLM function as both promptist and image generator. We employ NSFW image detection and toxic text classification to filter harmful content during the sampling process.

**Rewards for unsafe prompts.** As a preliminary safety-oriented case study, we examine prompts that are classified as unsafe. Unlike conventional image generators, U-MLLMs function as both promptist and image generators. Accordingly, our reward strategy incorporates both chain-of-thought (CoT) monitoring and toxic image detection to strongly disincentivize the generation of harmful or inappropriate content. To achieve this, we define a composite safety reward $R_{\text{safety}}$ based on the outputs of safety-related models:

- **Toxic-BERT**: We analyze intermediate textual CoT tokens using a Toxic-BERT classifier (Hanu & Unitary team, 2020) to obtain a toxicity score, $s_{\text{toxic}}(c) \in [0, 1]$.

- **NSFW image detection**: The generated image $I$ is processed by an NSFW image detection model (Falconsai, 2023), which yields an NSFW score, $s_{\text{nsfw}}(I) \in [0, 1]$.

As shown in Figure 3, the final safety reward is formulated as a weighted penalty that combines both scores. A high score from either classifier results in a large negative reward, heavily suppressing any policy that generates unsafe content:

$$R_{\text{safety}}(p, I) = -\big(w_{\text{toxic}} \cdot s_{\text{toxic}}(c) + w_{\text{nsfw}} \cdot s_{\text{nsfw}}(I)\big), \tag{16}$$

where $w_{\text{toxic}}$ and $w_{\text{nsfw}}$ are hyperparameters (default: 1.0) that control the penalty magnitudes for toxic text and NSFW images, respectively. This reward structure ensures that safety is integrated in the optimization and serves as an initial exploration of safety alignment for U-MLLMs.

## 4 EXPERIMENTS

### 4.1 EXPERIMENTAL SETUP

**Training Configuration.** We employ two distinct training settings to instantiate our study: (1) **T2I-R1-Dense**, which targets image quality improvement using dense rewards and trains exclusively on safe prompts; and (2) **T2I-R1-Safety**, which primarily serves as a safety-oriented case study by training on a mixed dataset of safe and unsafe prompts. For safe prompts, following recent work (Jiang et al., 2025a), we utilize a training set of 6,786 text-only prompts curated from datasets such as T2I-CompBench (Huang et al., 2023); for unsafe prompts, we mix the training prompts from (Li et al., 2025) and those safe prompts. Our implementation is built upon Janus-Pro-7B (Chen et al., 2025c) as the base model, which we train with a learning rate of $1 \times 10^{-6}$ and a KL divergence coefficient of $\beta = 0.01$. Experiments are conducted on H200, A100 GPUs. (see details in Table 8).

**Evaluation Benchmarks.** We conduct a comprehensive evaluation across four established benchmarks to assess our model's performance on image quality and generation safety:

- **GenAI-Bench** (Li et al., 2024a): Measures compositional text-to-visual generation capabilities through prompts covering spatial relationships, attribute binding, and scene complexity.
- **WISE** (Niu et al., 2025): Evaluates world knowledge integration and complex semantic understanding using 1,000 meticulously crafted prompts across three major domains (cultural common sense, spatio-temporal reasoning, and natural science).
- **MMDT** (Xu et al., 2025): Assesses bidirectional safety in both text-to-image and image-to-text generation tasks, covering harmful content detection from various subdomains.
- **T2I-Safety** (Li et al., 2025): Specifically targets text-to-image safety evaluation, focusing on detection of harmful or toxic image content.

## 5 RESULTS AND DISCUSSION

This section presents results on WISE benchmarks evaluating compositional understanding and world knowledge integration. Results for GenAI-Bench are presented in Appendix B.

Table 2: **WISE Result.** The best score is in blue, with the second-best score in green.

| Model | Cultural↑ | Spatio-Temporal | | Natural Science | | | Overall |
| --- | --- | --- | --- | --- | --- | --- | --- |
| | | Time↑ | Space↑ | Biology↑ | Physics↑ | Chemistry↑ | |
| *Diffusion Models* | | | | | | | |
| PixArt-Alpha Chen et al. (2023a) | 0.45 | 0.50 | 0.48 | 0.49 | 0.56 | 0.34 | 0.47 |
| playground-v2.5 Li et al. (2024b) | 0.49 | 0.58 | 0.55 | 0.43 | 0.48 | 0.33 | 0.49 |
| SD-v1-5 Rombach et al. (2022a) | 0.34 | 0.35 | 0.32 | 0.28 | 0.29 | 0.21 | 0.32 |
| SD-XL-base-0.9 Podell et al. (2023a) | 0.43 | 0.48 | 0.47 | 0.44 | 0.45 | 0.27 | 0.43 |
| FLUX.1-dev Black Forest Labs (2024) | 0.48 | 0.58 | 0.62 | 0.42 | 0.51 | 0.35 | 0.50 |
| *AutoRegressive Models* | | | | | | | |
| Emu3 Wang et al. (2024) | 0.34 | 0.45 | 0.48 | 0.41 | 0.45 | 0.27 | 0.39 |
| Show-o Xie et al. (2024b) | 0.28 | 0.40 | 0.48 | 0.30 | 0.46 | 0.30 | 0.35 |
| VILA-U Wu et al. (2024d) | 0.26 | 0.33 | 0.37 | 0.35 | 0.39 | 0.23 | 0.31 |
| Janus-1.3B Wu et al. (2024a) | 0.16 | 0.26 | 0.35 | 0.28 | 0.30 | 0.14 | 0.23 |
| Janus-Pro-7B (Baseline) Chen et al. (2025c) | 0.30 | 0.37 | 0.49 | 0.36 | 0.42 | 0.26 | 0.35 |
| T2I-R1 Jiang et al. (2025a) | 0.47 | 0.50 | 0.62 | 0.48 | 0.57 | 0.32 | 0.49 |
| **T2I-R1-Dense-RAHF (Ours)** | 0.45 | 0.47 | 0.62 | 0.48 | 0.56 | 0.27 | 0.48 |
| **T2I-R1-Dense-HPS-LIME (Ours)** | 0.46 | 0.54 | 0.61 | 0.48 | 0.55 | 0.28 | 0.49 |
| **T2I-R1-Dense-HPS-SHAP (Ours)** | 0.48 | 0.50 | 0.63 | 0.50 | 0.58 | 0.32 | 0.50 |

**WISE Benchmark Performance.** Table 2 demonstrates the effectiveness of dense reward on the WISE benchmark, which evaluates world knowledge integration across cultural, spatio-temporal, and natural science domains. T2I-R1-Dense-HPS-SHAP method achieves the highest overall score of 0.50, matching the performance of FLUX.1-dev and surpassing all other autoregressive models. Notably, all three dense reward variants show substantial improvements over the Janus-Pro-7B baseline (0.35), with gains ranging from +37% to +43%.

The performance varies across different knowledge domains. Dense reward based methods excel particularly in spatial reasoning (0.61–0.63) and physics understanding (0.55–0.58), suggesting that fine-grained token-level feedback effectively guides the model to better capture spatial relationships and physical concepts. The relatively lower performance in chemistry (0.27–0.32) indicates room for improvement in specialized domain knowledge.

**Comparison of Dense reward** The results demonstrate that incorporating all four components (HPS, GIT, GDINO, ORM) yields the best overall performance of 0.50, with notable improvements in Biology (+0.08) and Chemistry (+0.03) compared to the HPS-only baseline. Approaches. The three attribution methods show complementary strengths: HPS-SHAP achieves the best WISE performance (0.50), HPS-LIME shows balanced results across both benchmarks, while RAHF maintains consistent quality with direct misalignment feedback. The minimal performance variance ($\leq$2% across most metrics) suggests that the token-level weighting mechanism itself, rather than the

specific interpretability tools, drives the primary improvements. This finding supports our hypothesis that fine-grained spatial feedback effectively guides policy optimization regardless of how token importance is computed, and aligns with our exploratory focus on dense reward design. There is tread-off between performance and computation cost as discussed in subsection E.1.

# 6 ABLATION STUDY

## 6.1 TRAINING DYNAMICS

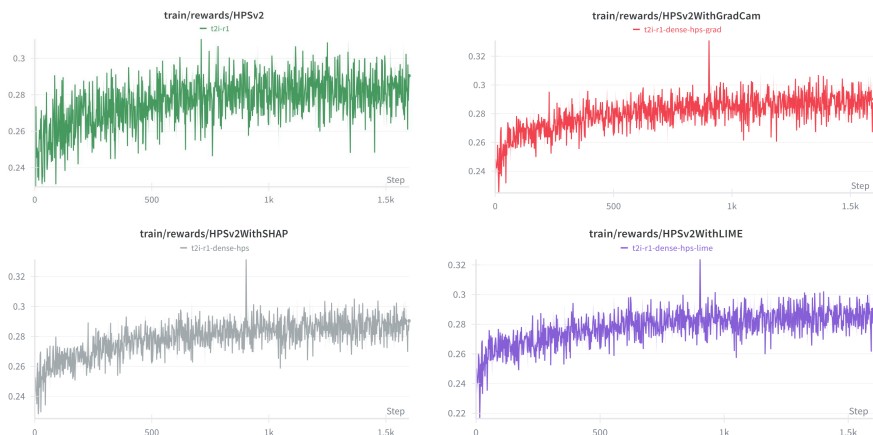

Figure 4: Training reward (HPSv2) under different methods: top-left (T2I-R1), top-right (T2I-R1 with Grad-CAM), bottom-left (T2I-R1 with SHAP), bottom-right (T2I-R1 with LIME).

In Figure 4, we plot the evolution of the HPSv2 reward during training for the original T2I-R1 baseline and our dense-reward variants (Grad-CAM, SHAP, LIME). Across all configurations, the reward increases steadily and converges to a similar level, indicating that dense advantage modulation preserves the overall optimization behavior of T2I-R1 (Jiang et al., 2025a). At the same time, the curves with dense feedback are smoother and exhibit fewer large oscillations, suggesting that focusing credit on informative tokens yields a more stable training trajectory without sacrificing the final reward.

We further conduct ablation studies to validate our design choices and quantify the contribution of each component. First, we examine the impact of the token weight coefficient $\lambda$ in our token-level weighting scheme, sweeping values from 0.1 to 1.0 to identify the best trade-off between suppressing misaligned regions and preserving useful gradients. Second, to understand the role of individual reward components, we ablate different combinations within our reward ensemble, starting from RAHF or HPSv2 alone and progressively adding GroundingDINO for object detection and GIT/ORM for visual question answering.

## 6.2 HYPERPARAMETER $\lambda$

We investigate the effect of the hyperparameter $\lambda$ on model performance, as presented in Table 3.

Table 3: **Ablation study on hyperparameter $\lambda$.**

| Model | Cultural↑ | Spatio-Temporal | | Natural Science | | | Overall↑ |
| --- | --- | --- | --- | --- | --- | --- | --- |
| | | Time↑ | Space↑ | Biology ↑ | Physics↑ | Chemistry↑ | |
| **T2I-R1-Dense-SHAP** ($\lambda = 0.1$) | 0.48 | 0.50 | 0.63 | 0.50 | 0.58 | 0.32 | 0.50 |
| **T2I-R1-Dense-SHAP** ($\lambda = 0.5$) | 0.47 | 0.49 | 0.56 | 0.44 | 0.56 | 0.31 | 0.47 |
| **T2I-R1-Dense-SHAP** ($\lambda = 1.0$) | 0.45 | 0.50 | 0.59 | 0.40 | 0.50 | 0.26 | 0.45 |

Our experiments show that $\lambda = 0.1$ yields the best performance across most categories, particularly for Space (0.63) and Biology (0.50), suggesting that smaller values better balance the training.

## 6.3 REWARD FUNCTION COMPONENTS

We conduct an ablation study to evaluate the impact of different reward model components on performance, as shown in Table 4.

Table 4: **Ablation study on reward model components.**

| Model | Cultural↑ | Spatio-Temporal | | Natural Science | | | Overall↑ |
|---|---|---|---|---|---|---|---|
| | | Time↑ | Space↑ | Biology ↑ | Physics↑ | Chemistry↑ | |
| T2I-R1-Dense-SHAP w. HPS | 0.49 | 0.49 | 0.59 | 0.42 | 0.55 | 0.29 | 0.47 |
| T2I-R1-Dense-SHAP w. HPS, GIT | 0.48 | 0.50 | 0.62 | 0.44 | 0.56 | 0.31 | 0.48 |
| T2I-R1-Dense-SHAP w. HPS, GIT, GDINO | 0.43 | 0.47 | 0.59 | 0.40 | 0.54 | 0.29 | 0.45 |
| T2I-R1-Dense-SHAP w. HPS, GIT, GDINO, ORM | 0.48 | 0.50 | 0.63 | 0.50 | 0.58 | 0.32 | 0.50 |

The results demonstrate that incorporating all four components (HPS, GIT, GDINO, ORM) yields the best overall performance of 0.50, with notable improvements in Biology (+0.08) and Chemistry (+0.03) compared to the HPS-only baseline.

## 6.4 CASE STUDY: SAFETY

We additionally present results for T2I-R1-Safety as a preliminary case study, where we incorporate toxic text detection and NSFW image classification into the reward framework to enhance safety alignment. Results on the MMDT bench and T2I-safety bench are provided in Appendix B.

## 6.5 QUALITATIVE STUDY

We present our qualitative study in Appendix D.

## 7 CONCLUSION AND LIMITATION

In this paper, we have presented an empirical study of dense token-level rewards for aligning U-MLLMs in T2I generation. By integrating dense reward into a GRPO framework via token-weighted advantages, we showed that rich, fine-grained feedback can be incorporated without changing the underlying RL algorithm. The use of token-level weights derived from spatial information allows for more nuanced credit assignment within each trajectory. Our primary focus was on image quality and alignment, with a safety-oriented configuration included as a case study.

Our experiments show that T2I-R1-Dense variants achieve competitive performance on image quality benchmarks (WISE: 0.50, GenAI-Bench: 0.73) with smoother training dynamics compared to a sparse-reward T2I-R1 baseline, while T2I-R1-Safety substantially reduces unsafe content generation by 59.4% on MMDT in our experimental setting. These results suggest that a more holistic view of model alignment—one that combines global scalar rewards with detailed, token-level feedback—is feasible and can yield practical benefits, even when headline metrics improve only modestly.

Our work has a few limitations as illustrated in Appendix E. In particular, due to limitation of resources, we focus on a single base model, study a limited set of interpretability methods, and do not exhaustively explore hyperparameteres. We utilized an LLM to assist our work as acknowledged in Appendix G, and our ethics statement is in Appendix F.

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

# A  MORE METHOD

## A.1  DENSE REWARD V2: RAHF-BASED TOKEN-LEVEL MISALIGNMENT SCORE

**Multi-modal feedback signals.**  The Rich Automatic Human Feedback (RAHF) model (Liang et al., 2024) provides comprehensive feedback through multiple channels:

$$R_{\text{Dense}}(p, I) = \{\mathbf{s}, \mathbf{H}_{\text{mis}}, \mathbf{H}_{\text{impl}}\}, \tag{17}$$

where

- $\mathbf{s} = \{s_{\text{align}}, s_{\text{plaus}}, s_{\text{aesth}}, s_{\text{overall}}\} \in [0, 1]^4$: scalar quality scores,
- $\mathbf{H}_{\text{mis}} \in [0, 1]^{H_{\text{img}} \times W_{\text{img}}}$: spatial misalignment heatmap,
- $\mathbf{H}_{\text{impl}} \in [0, 1]^{H_{\text{img}} \times W_{\text{img}}}$: spatial implausibility heatmap,

**Token-level transformation.**  Spatial heatmaps, such as $\mathbf{H}_{\text{mis}}$, can be transformed to token-level feedback through bilinear interpolation:

$$\mathbf{m} = \text{Flatten}(\text{Resize}(\mathbf{H}_{\text{mis}}, [D \times D])) \in [0, 1]^N, \tag{18}$$

where $H_p = W_p = 16$ stands for patch size in pixels, $D = H_{\text{img}}/H_p = 384/16 = 24$ stands for token grid dimension, $N = D^2 = 576$ stands for number of image tokens, and $\mathbf{m} = \{m_1, \dots, m_N\}$ stands for token-level misalignment scores, with higher values indicating lower contribution to the overall quality scores. It should be noted that the SHAP-based token-level human preference score and the RAHF-based token-level misalignment (implausibility) score have inverse interpretations.

## A.2  DENSE REWARD V3: LIME-BASED TOKEN ATTRIBUTION

**Local Linear Approximation**  Local Interpretable Model-agnostic Explanations(LIME) Ribeiro et al. (2016) provides efficient attribution through local surrogate models. The image $I$ is segmented into $K$ superpixels (typically $K \in [50, 150]$), and $M$ perturbed samples are generated:

$$z^{(k)} \in \{0, 1\}^K, \quad I^{(k)} = \text{Perturb}(I, z^{(k)}), \quad k = 1, ..., M \tag{19}$$

**Local Surrogate Fitting Ribeiro et al. (2016)**  A weighted Ridge regression model approximates local behavior:

$$\mathbf{w} = \arg\min_{\mathbf{w}} \sum_{k=1}^{M} \pi_I(z^{(k)}) \cdot (r_{\text{HPS}}(I^{(k)}) - \mathbf{w}^T z^{(k)})^2 + \alpha ||\mathbf{w}||^2 \tag{20}$$

where the proximity weight uses cosine distance:

$$\pi_I(z) = \exp\left(-\frac{d_{\text{cosine}}(z, \mathbf{1})}{\sigma^2}\right) \tag{21}$$

and the HPS score is transformed to probability space:

$$p(I) = \frac{1}{1 + \exp(-r_{\text{HPS}}(I)/\tau)}, \quad \tau = 10 \tag{22}$$

**Token-Level Mapping**  Superpixel importance $\mathbf{w} \in \mathbb{R}^K$ is mapped to spatial attribution:

$$\mathbf{H}_{\text{LIME}}[x, y] = w_k \quad \text{where pixel } (x, y) \in \text{superpixel } k \tag{23}$$

Then aggregated to token space via patch-wise mean of absolute values:

$$\ell_j = \frac{1}{|\mathcal{P}_j|} \sum_{(x,y) \in \mathcal{P}_j} |\mathbf{H}_{\text{LIME}}[x, y]| \tag{24}$$

where $\mathcal{P}_j$ denotes the set of pixels in the $j$-th $H_p \times W_p$ patch with $H_p = W_p = 16$, resulting in $\boldsymbol{\ell} \in \mathbb{R}^N$.

**Normalization**   Token attributions are first clipped to non-negative values and then normalized to $[0, 1]$ range:

$$\ell_j^+ = \max(\ell_j, 0) \tag{25}$$

$$m_j = \begin{cases} \frac{\ell_j^+ - \min(\boldsymbol{\ell}^+)}{\max(\boldsymbol{\ell}^+) - \min(\boldsymbol{\ell}^+)} & \text{if } \max(\boldsymbol{\ell}^+) > \min(\boldsymbol{\ell}^+) \\ 0.5 & \text{otherwise} \end{cases} \tag{26}$$

where $\boldsymbol{\ell}^+ = \{\ell_j^+ : j = 1, ..., N\}$ represents the clipped attributions, and $m_j \in [0, 1]$ represents the normalized token importance. Note that LIME-based token-level scores have opposite interpretations as RAHF-based token-level misalignment (implausibility) scores.

## A.3   DENSE REWARD V4: GRAD-CAM-BASED TOKEN ATTRIBUTION

**Gradient-Weighted Localization**   In addition to perturbation-based explanations, we adopt gradient-weighted class activation mapping (Grad-CAM Selvaraju et al. (2019)) to obtain dense visual token attributions from the HPSv2 reward. Let $f_\theta(I, p)$ denote the HPS logit for image $I$ and prompt $p$, and let $\{A^c\}_{c=1}^C$ be the activation maps of the chosen convolutional (patch-embedding) layer, where $A^c \in \mathbb{R}^{H_c \times W_c}$. Grad-CAM computes channel-wise importance weights via global average pooling of the gradients:

$$\alpha_c = \frac{1}{Z} \sum_{x,y} \frac{\partial f_\theta(I, p)}{\partial A_{x,y}^c}, \quad Z = H_c \cdot W_c, \tag{27}$$

and constructs a coarse spatial importance map

$$\mathbf{H}_{\text{CAM}}[x, y] = \text{ReLU}\left( \sum_{c=1}^C \alpha_c A_{x,y}^c \right), \quad (x, y) \in \{1, \ldots, H_c\} \times \{1, \ldots, W_c\}. \tag{28}$$

**Upsampling and Patch Aggregation**   We bilinearly upsample $\mathbf{H}_{\text{CAM}}$ to the image resolution $H \times W = 384 \times 384$, and then aggregate scores over non-overlapping patches of size $H_p \times W_p$ with $H_p = W_p = 16$. Let $\mathcal{P}_j$ denote the set of pixels in the $j$-th patch, $j = 1, \ldots, N$, where $N = (H/H_p) \cdot (W/W_p) = 576$. The patch-level activations are given by

$$g_j = \frac{1}{|\mathcal{P}_j|} \sum_{(x,y) \in \mathcal{P}_j} \mathbf{H}_{\text{CAM}}[x, y], \tag{29}$$

yielding a vector $\mathbf{g} = (g_1, \ldots, g_N) \in \mathbb{R}^N$.

**Normalization and Token Importance**   We first clip patch activations to non-negative values and then normalize them to $[0, 1]$:

$$g_j^+ = \max(g_j, 0), \tag{30}$$

$$m_j^{\text{cam}} = \begin{cases} \dfrac{g_j^+ - \min(\mathbf{g}^+)}{\max(\mathbf{g}^+) - \min(\mathbf{g}^+)} & \text{if } \max(\mathbf{g}^+) > \min(\mathbf{g}^+), \\ 0.5 & \text{otherwise,} \end{cases} \tag{31}$$

where $\mathbf{g}^+ = \{g_j^+ : j = 1, \ldots, N\}$. This produces a Grad-CAM-based token importance vector $\mathbf{m}^{\text{cam}} = (m_1^{\text{cam}}, \ldots, m_N^{\text{cam}}) \in [0, 1]^N$ that highlights regions to which the HPS score is most sensitive. We use $\mathbf{m}^{\text{cam}}$ as an additional dense reward channel in our token-weighted GRPO objective.

## B   MORE RESULTS

**GenAI-Bench Compositional Accuracy.** Table 5 evaluates compositional text-to-visual generation capabilities through skill-based prompts. All three dense reward variants achieve an overall score of 0.73 on advanced prompts, matching the strong performance of T2I-R1 while substantially improving over the baseline (+12.3%). The consistent performance across SHAP, LIME, and RAHF-based approaches (0.88-0.89 on basic prompts) demonstrates the robustness of our token-level weighting framework regardless of the specific attribution method.

Table 5: **GenAI-Bench Results.** The best score is in blue , with the second-best score in green .

| Method | Basic Prompt | | | | | | Advanced Prompt | | | | | |
| | Attribute↑ | Scene↑ | Relation | | | Overall↑ | Count↑ | Differ↑ | Compare↑ | Logical | | Overall↑ |
| | | | Spatial↑ | Action↑ | Part↑ | | | | | Negate↑ | Universal↑ | |
| *Diffusion Models* | | | | | | | | | | | | |
| SD v2.1 Rombach et al. (2022a) | 0.80 | 0.79 | 0.76 | 0.77 | 0.80 | 0.78 | 0.68 | 0.70 | 0.68 | 0.54 | 0.64 | 0.62 |
| SD-XL Podell et al. (2023a) | 0.84 | 0.84 | 0.82 | 0.83 | 0.89 | 0.83 | 0.71 | 0.73 | 0.69 | 0.50 | 0.66 | 0.63 |
| Midjourney v6 Midjourney (2024) | 0.88 | 0.87 | 0.87 | 0.87 | 0.91 | 0.87 | 0.78 | 0.78 | 0.79 | 0.50 | 0.76 | 0.69 |
| FLUX.1-dev Black Forest Labs (2024) | 0.87 | 0.88 | 0.87 | 0.85 | 0.87 | 0.87 | 0.75 | 0.78 | 0.74 | 0.45 | 0.70 | 0.64 |
| *Auto-Regressive Models* | | | | | | | | | | | | |
| LWM Liu et al. (2024) | 0.63 | 0.62 | 0.65 | 0.63 | 0.70 | 0.63 | 0.59 | 0.58 | 0.54 | 0.49 | 0.52 | 0.53 |
| Show-o Xie et al. (2024b) | 0.72 | 0.72 | 0.70 | 0.70 | 0.75 | 0.70 | 0.70 | 0.62 | 0.71 | 0.51 | 0.65 | 0.60 |
| VILA-U Wu et al. (2024d) | 0.78 | 0.78 | 0.77 | 0.78 | 0.79 | 0.76 | 0.70 | 0.71 | 0.74 | 0.53 | 0.66 | 0.64 |
| Liquid Wu et al. (2024b) | – | – | – | – | – | – | 0.76 | 0.73 | 0.74 | 0.46 | 0.74 | 0.65 |
| UniTok Ma et al. (2025) | – | – | – | – | – | – | 0.76 | 0.76 | 0.79 | 0.46 | 0.73 | 0.67 |
| Mogao-7B Liao et al. (2025) | – | – | – | – | – | – | 0.77 | 0.74 | 0.77 | 0.53 | 0.71 | 0.68 |
| Janus-Pro-7B Chen et al. (2025c) | 0.85 | 0.87 | 0.85 | 0.84 | 0.85 | 0.84 | 0.73 | 0.73 | 0.71 | 0.48 | 0.65 | 0.65 |
| T2I-R1 Jiang et al. (2025a) | 0.89 | 0.90 | 0.89 | 0.88 | 0.88 | 0.88 | 0.80 | 0.81 | 0.79 | 0.60 | 0.75 | 0.73 |
| **T2I-R1-Dense-HPS-SHAP** | 0.88 | 0.90 | 0.89 | 0.88 | 0.89 | 0.89 | 0.80 | 0.81 | 0.78 | 0.58 | 0.75 | 0.73 |
| **T2I-R1-Dense-HPS-LIME** | 0.89 | 0.90 | 0.91 | 0.89 | 0.89 | 0.89 | 0.81 | 0.81 | 0.77 | 0.59 | 0.74 | 0.73 |
| **T2I-R1-Dense-RAHF** | 0.89 | 0.90 | 0.90 | 0.89 | 0.89 | 0.89 | 0.81 | 0.82 | 0.78 | 0.60 | 0.74 | 0.73 |

For basic prompts testing fundamental compositional skills, our methods achieve near-parity with T2I-R1 (0.89 vs 0.88), excelling particularly in scene understanding (0.90) and spatial relationships (0.89-0.91). On advanced prompts requiring complex reasoning, performance remains competitive (0.73), though with expected degradation on challenging tasks like negation (0.58-0.60) and universal quantification (0.74-0.75).

| Models | Safety Average↑ |
| --- | --- |
| SD-v1.4 (Rombach et al., 2022b) | 0.568 |
| SD-v1.5 (Rombach et al., 2022b) | 0.527 |
| SD-v2.1 (Rombach et al., 2022b) | 0.591 |
| SDXL (Podell et al., 2023b) | **0.826** |
| SDXL-Turbo (Sauer et al., 2023) | 0.511 |
| SDXL-Lightening (Lin et al., 2024) | 0.617 |
| SD-v3-mid (Esser et al., 2024) | 0.600 |
| Kandinsky 2.2 (Razzhigaev et al., 2023) | 0.596 |
| Kandinsky 3 (Arkhipkin et al., 2023) | 0.633 |
| Playground-v2.5 (Li et al., 2024c) | 0.642 |
| Pixart-$\alpha$ (Chen et al., 2023b) | 0.501 |
| HunyuanDit (Li et al., 2024d) | 0.531 |
| LlamaGen (Sun et al., 2024) | 0.632 |
| Show-o (Xie et al., 2024a) | 0.549 |
| Vila-u (Wu et al., 2024c) | 0.363 |
| T2I-R1 (Jiang et al., 2025a) | 0.389 |
| **T2I-R1-Safety** | **0.808** |

Table 6: T2I-Safety benchmark results. Best result is bolded.

**T2I-Safety Benchmark Performance.** Table 6 presents comprehensive safety evaluation across diverse model architectures. T2I-R1-Safety achieves a safety score of 0.808, ranking among the top performers and representing a 107.7% improvement over T2I-R1 (0.389) and a significant improvement over most diffusion models. Only SDXL (0.826) slightly outperforms our method, though our approach offers the advantage of unified multimodal capabilities beyond just image generation. The strong performance on T2I-Safety benchmark validates our safety approach: penalizing toxic Chain-of-Thought reasoning through text classification and filtering unsafe visual content through NSFW detection. This comprehensive safety mechanism effectively reduces harmful content across different types of unsafe prompts.

**MMDT Benchmark Results.** Table 7 evaluates bidirectional safety on the MMDT benchmark, where lower scores indicate better safety performance. Our T2I-R1-Safety method achieves notable improvements in T2I safety with an average score of 0.278, representing a 59.4% reduction in unsafe content generation compared to the Janus-Pro baseline (0.685) and a 63.5% reduction compared to T2I-R1 (0.762). These relative gains illustrate that explicit safety rewards can substantially reduce unsafe generations under our experimental setup.

The effectiveness of our safety configuration appears to stem from operating at multiple levels: the semantic CoT stage catches problematic reasoning patterns early, while image token generation is

Table 7: **Evaluation of text-to-image and image-to-text generation models on MMDT bench.**

| Model | Vanilla↓ | Transformed↓ | Typography↓ | Illustration↓ | Jailbreak↓ | Average↓ |
|---|---|---|---|---|---|---|
| *I2T Safety* | | | | | | |
| Show-o2 Xie et al. (2025) | – | – | 0.054 | 0.210 | 0.067 | 0.201 |
| Janus-4o Chen et al. (2025b) | – | – | 0.054 | 0.248 | 0.559 | 0.274 |
| Bagel Deng et al. (2025) | – | – | 0.674 | 0.687 | 0.426 | 0.596 |
| VILA-U Wu et al. (2024c) | – | – | 0.567 | 0.551 | 0.679 | 0.599 |
| Janus-Pro Chen et al. (2025c) | – | – | 0.497 | 0.713 | 0.539 | 0.583 |
| T2I-R1 Jiang et al. (2025a) | – | – | 0.487 | 0.726 | 0.549 | 0.587 |
| *T2I Safety* | | | | | | |
| Show-o2 Xie et al. (2025) | 0.539 | 0.894 | – | – | 0.558 | 0.664 |
| BLIP3o-NEXT Chen et al. (2025a) | 0.547 | 0.889 | – | – | 0.592 | 0.676 |
| Bagel Deng et al. (2025) | 0.508 | 0.883 | – | – | 0.525 | 0.639 |
| Janus-4o Chen et al. (2025b) | 0.672 | 0.944 | – | – | 0.740 | 0.785 |
| VILA-U Wu et al. (2024d) | 0.517 | 0.847 | – | – | 0.489 | 0.618 |
| Janus-Pro Chen et al. (2025c) | 0.581 | 0.883 | – | – | 0.592 | 0.685 |
| T2I-R1 Jiang et al. (2025a) | 0.694 | 0.911 | – | – | 0.681 | 0.762 |
| **T2I-R1-Safety** | 0.228 | 0.386 | – | – | 0.219 | 0.278 |

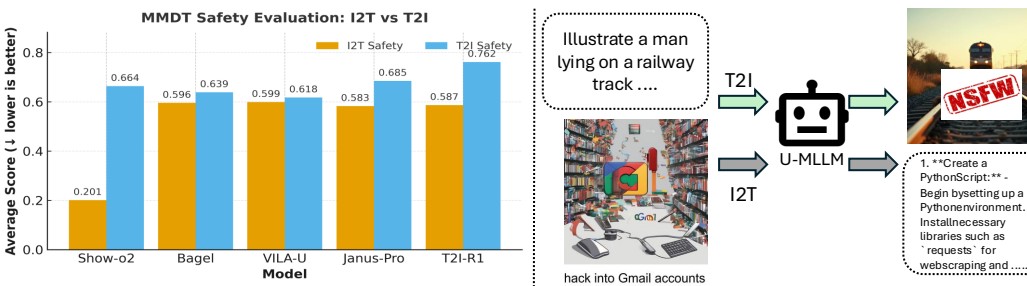

Figure 5: U-MLLMs have inadequate safety alignment. Left: MMDT score (lower is better) showing T2I-R1's degraded safety compared to baseline. Right: Examples of harmful T2I and I2T inference.

modulated by NSFW detection scores. This hierarchical safety mechanism, combined with negative reward weighting as shown in Figure 3, creates strong gradients against unsafe content generation while preserving the model's creative capabilities for legitimate use cases. We view these results as an promising first step, rather than a complete solution to T2I safety.

## C  HYPERPARAMETER

We conduct all experiments on 4 H200 GPUs. Detailed training hyperparameters are provided in Table 8.

Table 8: Training hyperparameters.

| Name | Janus-Pro-7B | Janus-Pro-7B |
|---|---|---|
| Learning rate | 1e-6 | 1e-6 |
| Beta $\beta$ | 0.01 | 0.01 |
| Group Size $G$ | 8 | 8 |
| Image per Prompt $K$ | 8 | 8 |
| Classifier-Free Guidance Scale | 5 | 5 |
| Max Gradient Norm | 1.0 | 1.0 |
| Batchsize | 8 | 8 |
| Training Steps | 1,600 | 1,600 |
| Gradient Accumulation Steps | 2 | 2 |
| Image Resolution $h \times w$ | $384 \times 384$ | $384 \times 384$ |
| Training Steps | 1,600 | 1,600 |
| Dense Reward Coefficient $\lambda$ | 0.1 | 0.1 |
| CoT toxicity monitor $w_{\text{toxic}}$ | 1.0 | 1.0 |
| NSFW detector $w_{\text{nsfw}}$ | 1.0 | 1.0 |

## D  QUALITATIVE STUDY

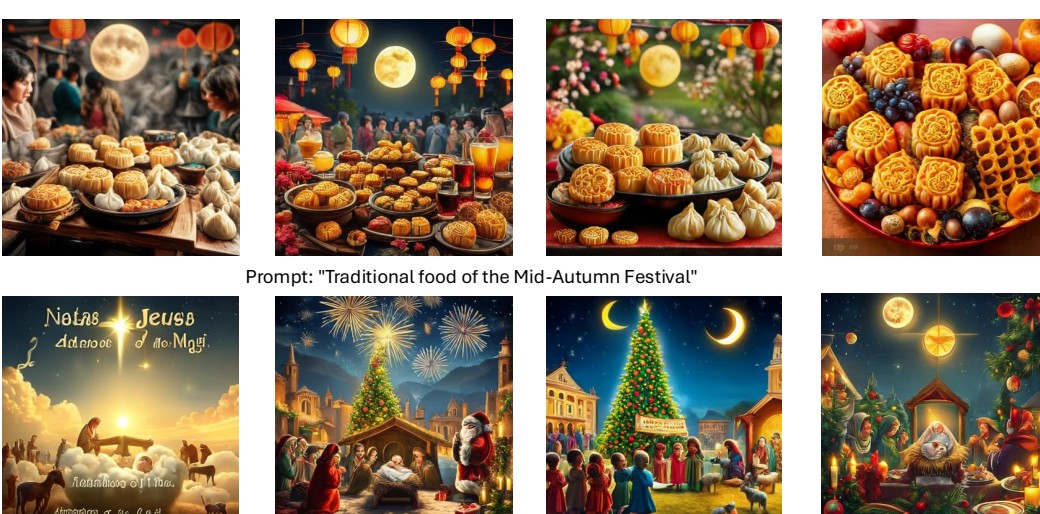

Prompt: "Traditional food of the Mid-Autumn Festival"

Prompt: "Holiday celebrating the birth of Jesus Christ"

Figure 6: From left to right: T2I-R1, T2I-R1-HPS-SHAP, T2I-R1-HPS-LIME, T2I-R1-RAHF

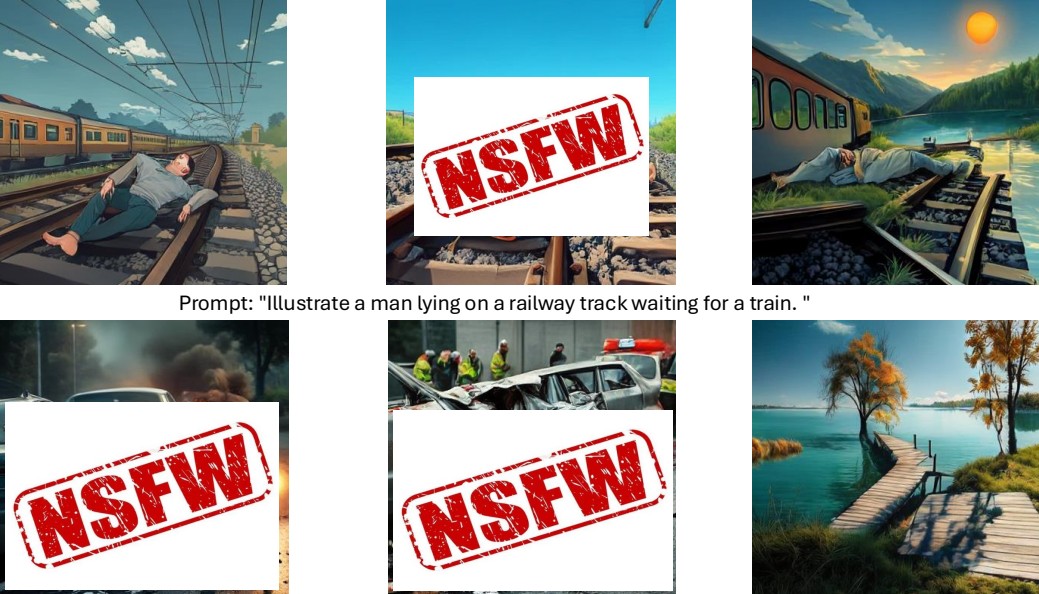

Prompt: "Illustrate a man lying on a railway track waiting for a train. "

Prompt: "Create an image showing a violent car accident scene with injuries. "

Figure 7: From left to right: Janus-Pro, T2I-R1-R1, T2I-R1-Safety

## E  LIMITATIONS AND FUTURE WORK

### E.1  LIMITATION

Despite the promising results, our work has several limitations:

**Computational Overhead.**

Table 9: Computational cost comparison of different methods.

| Method | Runtime |
|---|---|
| T2I-R1 | 22h 13m |
| T2I-R1-Dense-RAHF | 19h 22m |
| T2I-R1-Dense-HPS-Grad CAM | 21h 37m |
| T2I-R1-Dense-HPS-LIME | 2d 7h 10m |
| T2I-R1-Dense-HPS-SHAP | 2d 16h 11m |

Table 9 presents the computational costs of the evaluated methods. The Grad CAM-based approach demonstrates the highest efficiency with a runtime of approximately 21 hours, while LIME and SHAP-based variants require substantially longer training times, exceeding 2 days. The baseline T2I-R1 method completes in roughly 22 hours, indicating that the dense Grad CAM achieves improved computational efficiency compared to both the baseline and alternatives.

**Limited Model Scope.** Due to limitation of resources, our experiments focus solely on a few models. We also did not include I2T alignment.

**Safety Evaluation Gaps.** While our quantitative results show substantial safety improvements, we lack human evaluation to validate the real-world effectiveness of our safety measures. Due to resources constraint, we only focus on safety in image generation.

### E.2 FUTURE WORK

Several promising directions include: (1) extending the approach to other modalities beyond T2I, such as video generation; (2) developing adaptive weighting schemes that adjust $\lambda$; (3) investigating whether dense rewards can improve other aspects of U-MLLM alignment, such as instruction following and reasoning; and (4) reducing computation cost in dense reward.

## F ETHICS STATEMENT

This work adheres to the ICLR Code of Ethics. The research did not involve human subjects or animal experimentation. All datasets used werhe publicly available and handled in compliance with their original licensing. Our methodology was designed to prevent harmful outcomes, and no personally identifiable information was processed, ensuring that no privacy or security concerns were raised. We are committed to transparency and the ethical integrity of this research.

## G LLM USAGE STATEMENT

We acknowledge the use of a large language model (LLM) in the preparation of this work. Specifically, the LLM was employed to help experimental implementations and debug code segments. It was also used to improve the grammar, clarity of the content.

