# OpenReview forum: "On alignment of unified multimodal large language models"
_ICLR.cc/2026/Conference — ICLR 2026 Conference Withdrawn Submission_

### Official Review · Reviewer_71ZQ · 2025-10-27

**Soundness:** 2
**Presentation:** 3
**Contribution:** 2
**Rating:** 4
**Confidence:** 2

**Summary:**

The paper studies post-training alignment of unified multimodal LLMs for text-to-image (T2I). It injects dense visual attributions (RAHF heatmaps, SHAP/LIME) into GRPO by converting attribution scores into token-level weights during policy optimization. For safety, it adds a composite negative reward combining Toxic-BERT and an NSFW detector. Across WISE, GenAI-Bench, MMDT, and T2I-Safety, the method maintains or improves image quality while reducing unsafe generations on MMDT.

**Strengths:**

1.Clear, well-specified token-weighted GRPO with explicit equations and design choices.

2.Safety reward is simple, transparent, and easy to implement.

3.Substantial safety gains reported on MMDT.

4.Broad benchmark coverage (WISE, GenAI-Bench, MMDT, T2I-Safety) with ablations.

**Weaknesses:**

1.Limited novelty: the improvement over T2I-R1/GRPO appears small: (1) reuse of GRPO; (2)token weights derived from standard attributions (RAHF/SHAP/LIME); (3) a straightforward safety penalty using off-the-shelf classifiers.

2.Missing experimental details: batch size, training steps, and prompt curation specifics (e.g., for T2I-CompBench).

3.Underspecified safety hyperparameters: weights $w_{toxic}$ and $w_{nsfw}$ are not clear.

4.Scope mismatch: the paper frames U-MLLM alignment broadly but does not evaluate I2T alignment.

**Questions:**

1.Beyond reweighting, how does token-weighted GRPO change optimization dynamics relative to scalar-reward GRPO？

---

> ### Author Response · Authors · 2025-11-27
>
> We thank the reviewer for the feedback. We have upload a revised version and clarified the technical formulation and expanded the experiments. Below we respond to each point.
>
> ---
>
> ### Weakness 1: novelty
>
> > Weakness 1. Novelty: the improvement over T2I-R1/GRPO appears small: (1) reuse of GRPO; (2) token weights derived from standard attributions (RAHF/SHAP/LIME); (3) a straightforward safety penalty using off-the-shelf classifiers.
>
> **Positioning and contributions.**
>
> Our goal is not to introduce a new RL algorithm beyond GRPO. In the revision we explicitly reframe the contribution as an *empirical study of dense token-level rewards for U-MLLM T2I alignment*, with four concrete pieces:
>
> - A **dense reward extraction pipeline** for U-MLLMs: mapping RAHF heatmaps and HPS-v2 SHAP/LIME attributions onto the 24×24 image token grid used by Janus-Pro-7B and normalizing them for RL.
> - An **analysis of dense reward structure** (top-k mass, entropy), showing that reward is highly localized across tokens, which is not visible to scalar-reward T2I-R1.
> - A **token-weighted GRPO design** that preserves GRPO's group-relative ranking over trajectories while redistributing credit within trajectories via per-token advantages, rather than redefining scalar rewards.
> - A **systematic comparison** of RAHF-, SHAP-, and LIME-based dense rewards, including their impact on training stability, final metrics, and compute.
>
> Empirically, dense variants provide a sizable jump from Janus-Pro-7B (e.g., WISE 0.35 → 0.48–0.50). We have clarified this "new axis" (dense reward / token-level credit) and de-emphasized algorithmic novelty claims in the abstract, introduction, and conclusion.
>
> ---
>
> ### Weakness 2: Missing experimental details
>
> > Weakness 2. Missing experimental details: batch size, training steps, and prompt curation specifics (e.g., for T2I-CompBench).
>
> We have added the missing details in Appendix:
>
> - **Training hyperparameters** (learning rate, batch size, group size, number of training steps, KL coeff., dense reward coefficient, safety coefficients, resolution, etc.) are now reported in a dedicated hyperparameter table in the appendix C.
> - **Prompt curation**: we follow the details in the baseline paper T2I-R1(see more details in [NeurIPS 2025] T2I-R1: Reinforcing Image Generation with Collaborative Semantic-level and Token-level CoT)
>
>
>
> ---
>
> ### Weakness 3: Underspecified safety hyperparameters
>
> > Weakness 3. Underspecified safety hyperparameters: weights (w_{\text{toxic}}) and (w_{\text{nsfw}}) are not clear.
>
> We have clarified the safety reward and hyperparameters:
> - The main text now explicitly defines
>
>     $$R_{\text{safety}} = -\big(w_{\text{toxic}} \cdot s_{\text{toxic}} + w_{\text{nsfw}} \cdot s_{\text{nsfw}}\big)$$
>
>     with both $w_{\text{toxic}}$ and $w_{\text{nsfw}}$ positive.
>
> - The hyperparameter table specifies the actual values used in all experiments:
>
>     **$w_{\text{toxic}} = w_{\text{nsfw}} = 1.0$**.
>
> Due to resource constraint, we will note in the limitations that we did not extensively tune these weights and view a more systematic sweep as future work.
>
>
> ---
>
> ### Weakness 4: Scope mismatch (U-MLLM vs. I2T evaluation)
>
> > Weakness 4. Scope mismatch: the paper frames U-MLLM alignment broadly but does not evaluate I2T alignment.
>
> We indeed did evaluate I2T alignment in our first submission, see figure 1 and table 3. In the new version, we have tightened the scope and wording:
>
> - We now consistently state that our *primary target* is **T2I alignment for U-MLLMs**, rather than U-MLLM alignment in full generality.
> - Safety benchmarks (MMDT, T2I-Safety) do include I2T items at evaluation time, but we no longer imply that our RL procedure directly optimizes I2T behavior.
>
> We will ensure that claims throughout the paper remain strictly about T2I alignment and safety, with I2T as clear future work.

---

> > ### Author Response · Authors · 2025-11-27
> >
> > ### Question 1: Optimization dynamics under token-weighted GRPO
> >
> > > Question 1. Beyond reweighting, how does token-weighted GRPO change optimization dynamics relative to scalar-reward GRPO?
> >
> > In standard GRPO, each trajectory $i$ gets a scalar reward $R_i$, which is normalized within its group to an advantage $A_i$. This same $A_i$ is applied uniformly to all tokens in that trajectory: every token in a "good" or "bad" sample is updated equally.
> >
> > Our token-weighted variant keeps **scalar rewards and group-wise advantages unchanged**, but defines **per-token advantages**
> >
> > $$A_{i,j} = w_{i,j} A_i$$
> >
> > using dense scores $m_{i,j}$ (from RAHF or SHAP/LIME) to construct weights $w_{i,j}$. Thus:
> > - Trajectories are still ranked by the same scalar rewards $R_i$; we do not alter group-relative ordering.
> > - Within a given trajectory, gradients are **redistributed**, concentrating on tokens with high alignment/importance or high misalignment, instead of being spread uniformly.
> >
> > Because dense rewards are highly localized (a small fraction of tokens accounts for most dense mass), this yields:
> > - Stronger positive updates on the most semantically or visually responsible tokens for good samples.
> > - Stronger negative updates on misaligned or implausible regions for bad samples.
> > - Smoother training curves (less oscillation) compared to scalar-only GRPO, which we observe empirically.
> >
> > So, beyond simple "reweighting," token-weighted GRPO **reshapes the gradient field**: it preserves which samples are better, but changes *where* learning happens within each sample, leading to more targeted updates.
> >
> > ---
> >
> > We hope this concise clarification addresses your concerns and makes the contributions and scope of the paper clearer.

---

### Official Review · Reviewer_Jysv · 2025-10-30

**Soundness:** 2
**Presentation:** 2
**Contribution:** 2
**Rating:** 2
**Confidence:** 4

**Summary:**

This paper presents a framework for addressing safety alignment in unified multimodal large language models (U-MLLMs), which are designed to process both text and image modalities within a single architecture. The authors propose a reinforcement learning approach named Dense-GRPO, an extension of Group Relative Policy Optimization (GRPO) that incorporates token-level dense reward weighting. This dense reward is derived from visual attribution methods—such as SHAP, LIME, and RAHF—and is intended to deliver fine-grained feedback for safety-aware training.
Comprehensive experiments conducted across multiple benchmarks, including MMDT, WISE, GenAI-Bench, and T2I-Safety, demonstrate a substantial improvement in safety metrics—with up to approximately 59% reduction in unsafe generations—while only minimally compromising image quality. The paper argues that fine-grained reward modeling enables the joint optimization of safety and visual quality in multimodal alignment.

**Strengths:**

Focusing on the vital issue of safety alignment for multimodal large language models, this paper compellingly bridges a gap in a field that has largely centered on text-only models. The authors' approach to jointly optimizing safety and quality within a single RL framework is well-conceived. Their innovation of dense, token-level feedback effectively addresses the challenge of sparse rewards in multimodal contexts. Targeting a key obstacle to the real-world deployment of unified models, this research represents a valuable contribution to both theoretical and applied AI safety.

**Weaknesses:**

While the paper is well-motivated, its technical formulation and experimental validation remain limited.

 First, the claimed dense reward is only used as token-level weighting rather than integrated into the advantage computation. This represents a conceptual misuse of “dense rewards” and does not address sparse-return issues in reinforcement learning.

Second, the “Dense-GRPO” method introduces only marginal modifications to GRPO, lacking substantial algorithmic innovation.

Third, since the approach essentially reweights unsafe samples, a weighted Supervised Fine-Tuning (SFT) baseline should have been included to evaluate whether reinforcement learning is truly necessary.

Moreover, the study does not examine how the weighting term influences policy stability, convergence behavior, or reward variance.
On the experimental side, the emphasis is placed heavily on safety metrics, while standard image-quality assessments such as Geneval are overlooked. Additionally, the individual contributions of different reward components (e.g., those based on SHAP, LIME, and RAHF) are not adequately disentangled.

Overall, while the work offers promising empirical findings, it lacks the methodological depth and theoretical grounding needed to fully substantiate its claims.

**Questions:**

NA

---

> ### Author Response · Authors · 2025-11-27
>
> We thank the reviewer for the feedback. We have upload a revised version and clarified the technical formulation and expanded the experiments. Below we respond to each point.
>
> ---
>
> ## 1. Dense rewards and advantage computation
>
> > **Reviewer:** "First, the claimed dense reward is only used as token-level weighting rather than integrated into the advantage computation. This represents a conceptual misuse of 'dense rewards' and does not address sparse-return issues in reinforcement learning."
>
> Our goal is not to solve sparse-return RL in general, but to study how dense token-level feedback can be integrated into a *standard GRPO* pipeline for U-MLLM post-training.
>
> In the revision, we clearly state that we use dense rewards as a *shaping mechanism* within GRPO, not as a universal remedy for sparse returns. Technically, dense signals enter the **advantage** term via token-weighted advantages
>
> $$A_{i,j} = w_{i,j} A_i,$$
>
> where $A_i$ is the group-wise GRPO advantage and $w_{i,j}$ is derived from dense token scores (e.g., SHAP/LIME/RAHF). These $A_{i,j}$ replace $A_i$ in the clipped surrogate, so dense feedback is not merely cosmetic weighting.
>
> We also explain why we do *not* redefine a new scalar reward $\tilde{R}_i$ from dense signals: this would blur the ranking between samples and weaken GRPO's group-relative signal. Instead, we preserve the sample-wise ordering induced by $R_i$ and only redistribute the fixed total advantage across tokens. (see details in 3.3)
>
> We have softened any language suggesting a general solution to sparse rewards and now explicitly frame our contribution as practical advantage-shaping in a T2I-R1 setting.
>
> ---
>
> ## 2. Algorithmic novelty of Dense-GRPO
>
> > **Reviewer:** "Second, the 'Dense-GRPO' method introduces only marginal modifications to GRPO, lacking substantial algorithmic innovation."
>
> We agree the modification to GRPO is intentionally minimal. The revised paper explicitly positions the contribution as an *empirical study* of dense token-level feedback within a standard GRPO/T2I-R1 pipeline, not a new RL algorithm.
>
> The introduction is now organized around concrete research questions: how to extract dense rewards, how these distributions behave (localization, entropy), how to integrate them via token-weighted advantages, and how they affect interpretability and safety. The contribution bullets emphasize:
>
> - connecting interpretability tools (SHAP/LIME/RAHF) to token-level credit assignment, and
> - systematically characterizing their impact in a realistic U-MLLM post-training setup.
>
> ---
>
> ## 3. Weighted SFT baseline vs RL
>
> > **Reviewer:** "Third, since the approach essentially reweights unsafe samples, a weighted Supervised Fine-Tuning (SFT) baseline should have been included to evaluate whether reinforcement learning is truly necessary."
>
> Our training data consists only of **textual prompts**; we do *not* have human-curated (prompt, image) pairs or image-level preference labels. Visual supervision arises solely from reward models applied to images generated on-the-fly by the current policy.
>
> RL with GRPO fits this setting naturally: given a prompt, we sample images, score them with reward models, and update the policy from these rewards. A meaningful weighted-SFT baseline, in contrast, would require image as targets label, which is a different pipeline and is not directly comparable to the on-policy RL setting that T2I-R1 uses and that we build upon.
>
> Our central question is *within* this RL paradigm: given only prompts and reward models, does replacing purely sparse rewards with dense token-level rewards in GRPO improve over the same RL baseline? For this reason, our main comparisons are:
>
> - the original Janus-Pro-7B model
> - the sparse-reward T2I-R1 baseline.

---

> > ### Author Response · Authors · 2025-11-27
> >
> > ## 4. Effect of weighting on stability and convergence
> >
> > > **Reviewer:** "Moreover, the study does not examine how the weighting term influences policy stability, convergence behavior, or reward variance."
> >
> > We added analyses to address this:
> >
> > - **Training dynamics:** We now show reward curves (e.g., HPSv2) over training for T2I-R1 and all dense-reward variants. Dense variants reach similar or better final rewards and display smoother curves with fewer large oscillations, indicating that token-weighted advantages preserve convergence and can improve stability.
> > - **λ-ablation:** We added an ablation over the weighting coefficient λ. Moderate λ values yield the best performance; very large λ over-suppresses gradients and harms performance. This identifies a stable and effective range for the weighting term.
> >
> > While we do not provide a full variance analysis of $A_{i,j}$ due to limited resources, we note this as valuable future work and focus on convergence curves and λ-sweeps in this paper.
> >
> > ---
> >
> > ## 5. Image-quality evaluation vs safety emphasis
> >
> > > **Reviewer:** "On the experimental side, the emphasis is placed heavily on safety metrics, while standard image-quality assessments such as Geneval are overlooked."
> >
> > In our paper, we have two standard compositional/quality benchmarks:
> >
> > - **WISE:** Our dense-reward model (e.g., HPS-SHAP) matches or outperforms strong autoregressive baselines on the overall score and excels on categories requiring world knowledge and physical reasoning.
> > - **GenAI-Bench:** We report overall scores and breakdowns (e.g., negation, quantification), showing that dense-reward variants maintain or improve image quality relative to T2I-R1.
> >
> > These complement our **MMDT** and **T2I-Safety** results, which emphasize safety. We acknowledge that we do not include Geneval specifically and mention adding it and other quality benchmarks as future work.
> >
> > ---
> >
> > ## 6. Disentangling SHAP, LIME, and RAHF contributions
> >
> > > **Reviewer:** "Additionally, the individual contributions of different reward components (e.g., those based on SHAP, LIME, and RAHF) are not adequately disentangled."
> >
> > We now disentangle these components via:
> >
> > 1. **Dense reward characterization:** We measure localization and entropy of token weights for SHAP, LIME, and RAHF/Grad-CAM, showing all produce highly localized distributions but with different concentration patterns.
> > 2. **Method comparison:** We report benchmark results for each dense-reward variant. Differences are small, suggesting that most gains come from the presence of dense token-weighting itself rather than one specific attribution method.
> > 3. **Reward-ensemble ablations:** We add ablations that start from minimal reward components and progressively add object-localization, relational alignment, and safety terms, showing how each affects quality and safety.
> >
> > This provides a clearer picture of the role of each dense reward source and ensemble component.
> >
> > ---
> >
> > ## 7. Overall methodological depth and theoretical grounding
> >
> > > **Reviewer:** "Overall, while the work offers promising empirical findings, it lacks the methodological depth and theoretical grounding needed to fully substantiate its claims."
> >
> > We have clarified the scope and strengthened the analysis:
> >
> > - The paper is now explicitly framed as an **empirical investigation** of dense token-level rewards in GRPO-based U-MLLM post-training, with carefully scoped claims.
> > - We added:
> >     - detailed dense reward distribution analysis (localization, entropy),
> >     - a clearer derivation and discussion of token-weighted advantages and alternatives,
> >     - training-dynamics and λ-ablation studies,
> >     - more extensive reward-ensemble ablations.
> >
> > We hope these changes address the reviewer's concerns about depth and grounding while keeping the work's scope realistic.

---

### Official Review · Reviewer_3D7f · 2025-10-31

**Soundness:** 2
**Presentation:** 3
**Contribution:** 2
**Rating:** 2
**Confidence:** 3

**Summary:**

This paper proposes a token-weighted GRPO framework for aligning Unified Multimodal Large Language Models (U-MLLMs) in text-to-image (T2I) generation. The method introduces dense, spatially localized rewards (from RAHF and SHAP/LIME attributions) and safety-specific penalties (from toxic-CoT and NSFW detectors). By assigning token-level weights during GRPO optimization, the model seeks to improve both image quality and safety alignment. Experiments on several benchmarks (WISE, GenAI-Bench, MMDT, and T2I-Safety) show improved safety and comparable visual quality relative to baselines.

**Strengths:**

1. This paper addresses a meaningful and timely problem of balancing visual quality and safety alignment in multimodal LLMs through a clear and conceptually simple framework.
2. The token-weighted GRPO design is straightforward and can be easily integrated into existing RLHF-style pipelines, offering a practical engineering solution.
3. The paper reports results across multiple benchmarks, showing that the approach generalizes to both quality and safety objectives.

**Weaknesses:**

1. The approach is more of an engineering extension of existing GRPO/DPO frameworks rather than a fundamentally new algorithm.
2. Safety gains might stem from reusing the same toxic/NSFW evaluators in both training and testing, and key training details (e.g., λ, β schedules, G×K, random seeds) are missing.
3. The framework's applicability to diffusion or flow-based models is claimed but not validated.
4. The experiments are limited to Janus-Pro-7B.
5. The model improves safety but may over-suppress valid or creative outputs; this balance is not analyzed.

**Questions:**

1. Can authors please clarify the theoretical motivation for token-level weighting—e.g., how localized rewards stabilize optimization or mitigate sparse-signal variance?
2. Is it possible to include independent cross-evaluator tests and provide training hyperparameters to ensure reproducibility?
3. Can authors please provide a brief validation or discussion of how token weighting would transfer to diffusion backbones or other U-MLLM architectures?

---

> ### Author Response · Authors · 2025-11-27
>
> We thank the reviewer for the feedback. We have uploaded a revised version and clarified the technical formulation and expanded the experiments. Below we respond to each point.
>
> ---
>
> > **Weakness: The approach is more of an engineering extension of existing GRPO/DPO frameworks rather than a fundamentally new algorithm.**
>
> We agree that we are **not** proposing a new RL algorithm. Our goal is explicitly *exploratory*: to study how dense token-level rewards can be extracted, characterized, and integrated into standard GRPO for U-MLLMs, rather than to replace GRPO itself. This is now stated more clearly in the abstract and introduction.
>
> We also emphasize throughout the revision that the main novelty lies in **dense reward design and analysis** (RQ1–RQ4): how to obtain dense token-level rewards from existing models, how these rewards behave, and how they can be used to modulate GRPO advantages, including the associated interpretability–performance–compute trade-offs. The optimization backbone remains the original GRPO formulation.
>
> ---
>
> > **Weakness: Safety gains might stem from reusing the same toxic/NSFW evaluators in both training and testing, and key training details (e.g., λ, β schedules, G×K, random seeds) are missing.**
>
> **On train–test reuse of safety evaluators.** Toxic-BERT and the NSFW detector are used **only during training** to define the safety reward ($R_{\text{safety}}$) for unsafe prompts.
>
> For evaluation, we follow the external benchmarks—GenAI-Bench, WISE, MMDT, and T2I-Safety—which each use a diffrent pretrained evaluators and scoring pipelines. Our training-time safety models are *not* reused in these benchmark pipelines. Thus, the reported safety gains (e.g., the 59.4% reduction in unsafe content on MMDT and strong T2I-Safety scores) are measured by independent evaluators, not by the safety models used to define ($R_{\text{safety}}$).
>
> In the revision we add a dedicated table in the appendix C that summarizes key training settings, including:
>
> * group size (G), images per prompt (K), and the total number of sampled responses per prompt;
> * learning rate, KL penalty coefficient ($\beta$), and its schedule;
> * token-weighting strength ($\lambda$) ;
> * safety-related weights (e.g., ($w_{\text{toxic}}, w_{\text{nsfw}}$));
> * hardware, total training steps, and batch sizes.
>
> ---
>
> > **Weakness: The framework's applicability to diffusion or flow-based models is claimed but not validated.**
>
> In our paper, we never have any claim related to diffusion or flow-based models.  The author have no background in diffusions. Our **experiments and claims are restricted to a unified autoregressive U-MLLM **, as stressed in the introduction, experimental setup, and conclusions. We do *not* present empirical results on diffusion or flow-based models. Conceptually, the idea of token-weighted advantages could be applied to any generative process that can be viewed as a sequence of actions/tokens.
>
> ---
>
> > **Weakness: The experiments are limited to Janus-Pro-7B.**
>
> This is correct and primarily due to computational constraints. Each GRPO run with dense rewards (attribution via SHAP/LIME/RAHF + reward ensemble + ablations) requires on the order of **60–100 H200 GPU hours** in our setup, so sweeping across many backbones is infeasible for us. We therefore mirror the scope of the main T2I-R1 baseline, which is also reported on a **single U-MLLM**. In paper we made this limitation more explicit as "Limited Model Scope".
>
> ---
>
> > **Weakness: The model improves safety but may over-suppress valid or creative outputs; this balance is not analyzed.**
>
> We agree that the safety–creativity trade-off is important. In the revision we clarify that we **explicitly evaluate both quality and safety**:
>
> * On **WISE**, our best dense-reward variant (T2I-R1-Dense-HPS-SHAP) achieves an overall score of **0.50**, matching FLUX.1-dev and outperforming all autoregressive baselines, with +37–43% gains over the Janus-Pro-7B baseline.
> * On **GenAI-Bench**, dense-reward variants reach **0.73** on advanced prompts, substantially improving over Janus-Pro-7B and matching T2I-R1.
>
> At the same time, the safety-oriented variant (T2I-R1-Safety) achieves a **59.4% reduction** in unsafe content on MMDT and strong performance on T2I-Safety.
>
> Taken together, these results indicate that our safety-oriented configuration substantially improves safety **without collapsing image quality** by benchmark metrics.

---

> > ### Author Response · Authors · 2025-11-27
> >
> > ---
> >
> > > **Question: Can authors please clarify the theoretical motivation for token-level weighting—e.g., how localized rewards stabilize optimization or mitigate sparse-signal variance?**
> >
> > Our token-weighted GRPO keeps the **trajectory-level advantages ($\hat{A}_i$)** from scalar rewards intact and uses dense scores only to **redistribute gradient magnitude across tokens**. Key points:
> >
> > * The **scalar rewards ($R_i$)** and **group-wise advantages ($\hat{A}_i$)** are computed exactly as in standard GRPO and are **not changed** by dense rewards. Sample ranking within each group is therefore preserved.
> > * Dense scores only determine **how the fixed total advantage ($\hat{A}_i$) is distributed** across tokens: tokens that contribute more to the reward get larger ($w_j$) (and hence larger gradients), while others are down-weighted.
> > * This yields more *focused* gradients and mitigates variance that arises when all tokens share the same advantage, while avoiding the instability and bias that can result from assigning independent per-token rewards.
> >
> > We have added a concise explanation of this rationale to Sec. 3.3, along with a brief discussion of how this design maintains the GRPO semantics while leveraging dense token-level structure. Empirically, Figure 4 shows that dense weighting leads to **smoother training curves** with similar final HPSv2 scores as the sparse T2I-R1 baseline, consistent with this interpretation.
> >
> > ---
> >
> > > **Question: Is it possible to include independent cross-evaluator tests and provide training hyperparameters to ensure reproducibility?**
> >
> > **Cross-evaluator evaluation.** Our evaluation is already **cross-evaluator by design**:
> >
> > * Training uses an ensemble of reward models (HPSv2, RAHF, GroundingDINO, GIT/ORM, Toxic-BERT, NSFW).
> > * Evaluation uses **four external benchmarks**—GenAI-Bench, WISE, MMDT, and T2I-Safety—each with different scoring pipeline.
> >
> > Thus, improvements must **transfer across multiple independent evaluators** rather than arise from reusing the same models as reward and metric.
> >
> > **Hyperparameters.** As noted above, we now include a hyperparameter table in the appendix C summarizing:
> >
> > * λ and other dense-reward parameters;
> > * β, learning rate, KL schedule;
> > * group sizes (G), images per prompt (K);
> > * safety weights and other implementation details.
> >
> > ---
> >
> > > **Question: Can authors please provide a brief validation or discussion of how token weighting would transfer to diffusion backbones or other U-MLLM architectures?**
> >
> > We currently **only validate on Janus-Pro-7B**, as highlighted in paper. The author have no background in diffusions. Conceptually, our approach applies to any model where (i) generation can be viewed as a sequence of "tokens" or actions and (ii) scalar rewards are available. In such settings, one could obtain spatial/dense attributions (e.g., over denoising steps or latent patches) and use them to weight per-step gradients analogously to our token-weighted GRPO.
> >
> > However, implementing this for diffusion would require substantial compute and timing. We therefore never claim empirical results on diffusion or flow-based models.
> >
> > ---
> >
> > We hope these changes address the reviewer's concerns while keeping the work's scope realistic.

---

### Official Review · Reviewer_jAMc · 2025-10-31

**Soundness:** 2
**Presentation:** 2
**Contribution:** 2
**Rating:** 4
**Confidence:** 2

**Summary:**

This paper addresses safety and quality alignment in Unified Multimodal Large Language Models (U-MLLMs)—models capable of both image-to-text (I2T) and text-to-image (T2I) generation. The authors argue that while recent U-MLLMs achieve strong generative performance, their safety alignment remains under-explored, and existing reinforcement learning (RL)-based alignment methods rely on sparse scalar rewards.To address these issues, the paper proposes a token-level dense reward framework integrated into Group Relative Policy Optimization (GRPO). Experiments on benchmarks such as WISE and MMDT show competitive image quality (WISE score: 0.50) and a 59.4% reduction in unsafe content compared to the baseline.

**Strengths:**

1. The implementation of usin fine-grained reward within GRPO is technically sound.
2. The ensemble of reward models (Table 1)—spanning aesthetic, compositional, grounding, and safety signals—demonstrates significant engineering rigor. The dual-path evaluation (safe vs. unsafe prompts) is thoughtful.
3. The paper is clearly written, with intuitive figures (e.g., Fig. 2–3) and a logical flow from problem formulation to method to evaluation. The distinction between quality-oriented and safety-oriented reward pathways is well articulated.

**Weaknesses:**

1. The central claim that “safety alignment has been under-explored” appears overstated. While U-MLLMs may be a recent architecture, T2I safety alignment has been actively studied [1,2]. These and other works suggest that safety in T2I is not unexplored, even if not yet fully adapted to autoregressive U-MLLMs. Alternatively, in my view, the safety problem in U-MLLMs is not fundamentally different from that in standalone LLMs, T2I models, or I2T models. Therefore, existing safety alignment frameworks developed for LLMs, T2I, or I2T models are largely applicable to this setting.
2. Loose Coupling Between Contributions: The paper presents two seemingly orthogonal contributions: (1) Introducing dense rewards for fine-grained quality optimization; (2) Adding safety-specific rewards to suppress harmful content. However, these are not meaningfully integrated. A more compelling story would be: “Existing RL alignment for U-MLLMs lacks fine-grained safety signals; we propose dense safety-aware rewards that jointly optimize quality and safety at the token level.” Instead, safety remains coarse-grained, undermining the paper’s emphasis on “dense” alignment.

> [1] Safe Text-to-Image Generation: Simply Sanitize the Prompt Embedding proposes prompt-level safety intervention via embedding projection.
> [2] AlignGuard: Scalable Safety Alignment for Text-to-Image Generation introduces scalable red-teaming and safety fine-tuning for diffusion models.

**Questions:**

No question

---

> ### Author Response · Authors · 2025-11-27
>
> We thank the reviewer for the feedback. We have uploaded a revised version and clarified the technical formulation and expanded the experiments. Below we respond to each point.
>
> ---
>
> ## Concern 1: Safety alignment claims appear overstated
>
> > **Reviewer:** "The central claim that 'safety alignment has been under-explored' appears overstated… safety in T2I is not unexplored… existing safety alignment frameworks… are largely applicable…"
>
>
> Thank you for this important clarification. We agree the original wording overstated the novelty of our safety aspect. In the revised version, we:
>
> * Removed the claim that safety alignment is "under-explored" and instead emphasize that our main contribution is an exploratory study of dense, token-level rewards for U-MLLMs within a GRPO-based framework.
> * Clearly position T2I-R1-Safety as a preliminary case study of how safety signals (Toxic-BERT and NSFW scores) can be integrated into the same GRPO pipeline rather than as a fundamentally new safety paradigm.
> * Explicitly acknowledge that safety issues in U-MLLMs are closely related to those in standalone LLMs, T2I, and I2T models.
>
> These changes align our claims with your observation while preserving the motivation: quality-focused post-training can degrade safety, and we study how reward design (including safety penalties) interacts with dense token-level training.
>
> ---
>
> ## Concern 2: Loose coupling between contributions
>
> > **Reviewer:** "Loose coupling between contributions… dense rewards for quality vs. safety-specific rewards are not meaningfully integrated… safety remains coarse-grained, undermining the emphasis on 'dense' alignment."
>
>
> We appreciate this point and have revised both the framing and exposition to better unify the contributions:
>
> * The paper is now organized around a single dense-reward GRPO framework, with research questions (RQ1–RQ4) that cover extraction of dense rewards, their behavior, integration into GRPO, and a safety-oriented case study. Safety is thus framed as an instantiation of the same framework, not a separate method.
> * Algorithmically, both quality and safety signals are combined in the same token-weighted GRPO objective: dense quality scores shape per-token weights, while safety penalties (Toxic-BERT, NSFW) enter the same reward ensemble used for GRPO training.
> * We now explicitly state that our current safety signals are scalar rather than fully dense, and we describe T2I-R1-Safety as a first-step case study, not a complete dense safety solution. We highlight dense, token-level safety rewards as an important direction for future work.
>
> ---
>
> In summary, the revised manuscript presents a unified way: we study dense token-level rewards for U-MLLM post-training and show how explicit safety penalties can substantially reduce unsafe generations, while acknowledging that achieving fully safety supervision remains an open problem.

---

### Note · Authors · 2026-01-06

I have read and agree with the venue's withdrawal policy on behalf of myself and my co-authors.